# Cytokinin Signaling and De Novo Shoot Organogenesis

**DOI:** 10.3390/genes12020265

**Published:** 2021-02-12

**Authors:** Katarzyna Hnatuszko-Konka, Aneta Gerszberg, Izabela Weremczuk-Jeżyna, Izabela Grzegorczyk-Karolak

**Affiliations:** 1Department of Molecular Biotechnology and Genetics, Faculty of Biology and Environmental Protection, University of Lodz, Banacha 12/16, 90-237 Lodz, Poland; aneta.gerszberg@biol.uni.lodz.pl; 2Department of Biology and Pharmaceutical Botany, Medical University of Lodz, Muszynskiego 1, 90-151 Lodz, Poland; izabela.weremczuk-jezyna@umed.lodz.pl (I.W.-J.); izabela.grzegorczyk@umed.lodz.pl (I.G.-K.)

**Keywords:** cytokinins, shoot induction, regeneration, gene regulation, CKs receptor, two-component system

## Abstract

The ability to restore or replace injured tissues can be undoubtedly named among the most spectacular achievements of plant organisms. One of such regeneration pathways is organogenesis, the formation of individual organs from nonmeristematic tissue sections. The process can be triggered in vitro by incubation on medium supplemented with phytohormones. Cytokinins are a class of phytohormones demonstrating pleiotropic effects and a powerful network of molecular interactions. The present study reviews existing knowledge on the possible sequence of molecular and genetic events behind de novo shoot organogenesis initiated by cytokinins. Overall, the review aims to collect reactions encompassed by cytokinin primary responses, starting from phytohormone perception by the dedicated receptors, to transcriptional reprogramming of cell fate by the last module of multistep-phosphorelays. It also includes a brief reminder of other control mechanisms, such as epigenetic reprogramming.

## 1. Introduction

Being sessile organisms, plants have developed and mastered the art of adaptation. The ability to restore or replace injured tissues and structures undoubtedly can be named among their most spectacular achievements. This remarkable plasticity of the developmental and regenerative route is the function of regaining cellular totipotency [1]. Following the definitions by Gamborg and Phillips [2], regeneration can occur by either organogenesis as the formation of individual organs (e.g., shoots or roots), or somatic embryogenesis as the formation of a bipolar structure containing both shoot and root meristems, which develops in a manner similar to zygotic embryos. Both organogenesis and somatic embryogenesis are general terms that include direct (plant regeneration can be obtained from nonmeristematic tissue sections—adventitious origin) and indirect (plant regeneration can be obtained from callus and cell cultures—de novo origin) processes by which plants can respond to events of loss or damage of the body structures caused by various types of environmental stressors [2,3]. The most spectacular extreme of this plasticity includes the ability to spawn de novo meristems and organs from differentiated tissues, sometimes leading to a complete organ conversion following an identity switch by the cell.

This regenerative potential was exploited in many biotechnological (e.g., pharmaceutical or agricultural) practices, including the production of transgenic plants. The number of reported in vitro regeneration protocols from a wide range of source explants is vast and covers all known modes. However, behind all these events there are extremely complex molecular regulation and genetic mechanisms responsible for cell/tissue fate. Initial cues elicit transcriptional cascades and amend concentration and distribution of endogenous plant hormones. In laboratory conditions, such processes are triggered by wounding, incubation on the appropriate medium or supplementation with phytohormones, usually auxins and cytokinins. Consequently, the inductive signals are perceived by plant cells and modify ongoing metabolic and genetic settings, and the mode of reprogramming is started. However, precise knowledge on how the stimuli modulate the developmental pattern and how they point out the definite, as well as the optimal pathway of regeneration, is still to be acquired.

Cytokinins (CKs) represent one of the most important groups of plant hormones that can be used to exogenously provoke in vitro responses [4]. Indeed, cytokinins, discovered in the 1950s (isolation and characterization of kinetin in 1955), due to the wide range of their activities, are included in most publications on tissue culture [5,6]. These derivatives of N6-substituted adenine with isoprenoid side chains take part in nearly all aspects of the growth and development of plants (e.g., meristem maintenance, cell division, senescence of leaves, nodule formation, sink–source interaction, biotic and abiotic stress response and induction of plant immunity) [7,8,9]. The structures of example cytokinins are presented in Figure 1. Their exogenous application triggers inner regeneration protocols in many species. They are present in whole plant organisms and are synthesized in various types of cells in shoot and root tissues [10]. Moreover, being one of the key players of in vitro responses, they induce and/or reinforced proliferation in chosen cells. Hence, the set of summary data on the possible sequence of molecular and genetic events behind de novo shoot organogenesis (DNSO) initiated mostly by cytokinins is herein presented. The aspects discussed comprise different levels of regulation of cytokinin-related genes in shoot development, including epigenetic reprogramming. The regulation of DNSO occurs via the activation of key transcriptional regulators, dynamic changes in gene expression and impressive phytohormone crosstalk. However, chromatin availability, perhaps of a minor function here, is less frequently presented.

## 2. De Novo Shoot Organogenesis

De novo shoot organogenesis is the ability to perform a new complete process of postembryonic shoot formation. Typically, only a group of explant cells displays responsiveness to inductive signals during organogenesis as well as somatic embryogenesis from mature plant organs. A routine laboratory scheme of in vitro shoot regeneration may include using an auxin-enriched medium for the explant preculture to induce callus generation and subsequent cytokinin-dependent shoot induction. Thus, although this analysis was to focus on cytokinin-triggered events, one just cannot ignore an auxin-based medium intermediate. The golden hormonal regeneration pattern by Skoog and Miller (assuming the influence of mutual proportions of auxins and cytokinins) remains the guiding determinant of the in vitro fate of the explant [11]. A high cytokinin to auxin ratio stimulates shoot formation, while roots are formed when the ratio is low. This reciprocal connection between root and shoot formation scenarios is multilevel and clearly visible when the molecular background of shoot regeneration is analyzed. Intriguingly, it was demonstrated that in de novo shoot indirect organogenesis, the callus tissue corresponds to the tip of the root meristem. Such similarity is observed even if the callus originates from aerial plant organs (cotyledons, petals). Hence, the opinion that callus tissue is formed by simple reprogramming backward to an undifferentiated condition, becoming a mass of unorganized cells, is not so obvious. It was suggested that within a plant explant there can be other kinds of pre-existing, stem-like cells that are able to proliferate generating callus [12]. Strikingly, Atta et al. [13] showed that root and shoot material formed a callus tissue from pericycle cells neighboring the xylem poles when incubated on CIM (callus-inducing medium). A subsequent histological analysis found that the callus initially possessed ordered structures corresponding to the primordia of lateral roots [13]. A more detailed transcriptome analysis of callus-forming cells at the stage of differentiation confirmed the correspondence between profiles of the genetic expression of callus and of the root meristem [14,15]. This compelling evidence indicates that at least indirect shoot organogenesis engages a genetic pathway characteristic of lateral root induction. Despite the fact that, at the genetic level during shoot direct organogenesis, its formation pathway resembles that of the root, the precise character of this relationship is still to be specified.

Nevertheless, direct protocols to form shoots from explants are used just as often as the above ones, both resulting in two universal, to a certain extent, morphological steps: acquisition of pluripotency (as the cellular potential to beget different sorts of cells) and nonembryonic shoot organogenesis. Since a great number of excellent reviews on shoot regeneration has been published, the number of stages is higher depending on the details covered. According to some scientists, there are three general steps (given the exogenous plant hormones required and their role) including:cell dedifferentiation, as a consequence of the acquisition of morphogenic competence;response to exogenous plant hormones, understood as a determination of competent cells to form a shoot (induction);phytohormone-independent organ morphogenesis [16].

Reviewing the process of DNSO from calli Shin et al. [15] resulted in proposing a more detailed distinction. There, the four-phased DNSO course consisting of pluripotency acquisition, formation of the shoot promeristem, establishment of the confined shoot progenitor and shoot outgrowth, was discussed [15]. Nevertheless, no matter what theoretical division is made, recently, compelling evidence has shown that the regenerative pathway requires repression in the absence of the triggering factors. According to Ikeuchi et al. [3], a great number of seed plants developing different tissues has worked out mechanisms to constrain the regenerative capacity of their somatic cells. This continuous repression may ensure the maintenance of the operative integrity and allow for standard development [3]. Although it did not find an unequivocal confirmation in all publications (and it probably presumes a rather partial repression of chosen regeneration players), this hypothesis seems to illustrate that cellular identity is less fixed than initially assumed. Moreover, such repression would cast doubt on the understanding and character of pluripotency acquisition. Thus, it would seem more eligible to talk about pluripotency triggering or induction (understood as an already existing state) than about pluripotency acquisition. However, it depends on the level, which the regeneration is repressed at.

## 3. Cytokinin Signaling/Regulation

De novo shoot regeneration definitely requires cell proliferation involving the activation of the mitotic division. As has already been mentioned, cytokinins are a class of phytohormones with a wide spectrum of functions throughout the life cycle of plants, including cell division. Cytokinins influence the competent cells of the DNSO system leading to the production of a cell mass and cell fate conversion. This process involves cytokinin perception and its further signaling. It has been shown that a great part of this signaling occurs through the fine-tuned regulation of gene expression. It seems rather unlikely to succeed in isolating genetic or molecular players that are responsible only for shoot organogenesis. However, since the discovery of cytokinins, much work has been done to determine the biosynthetic enzymes, phytohormone receptors and their signal transduction pathways at various levels. The research on the regulation of gene expression included both the identification of the *cis*-elements in the nucleotide sequence and the screening of the trans factors involved. Brenner et al. [17] pointed out several milestones defined while investigating the cytokinin-dependent gene regulation: (i) finding of type-A Response Regulator (type-A RR) genes, considered to be the prime cytokinin response genes; (ii) discovery of type-B Response Regulators (type-B RR) that are Transcription Factors (TF) acting as mediators of the cytokinin response; (iii) recognition of a *cis*-acting cytokinin response sequences; (iv) dedicated genome analysis of the *Arabidopsis* transcriptome and finally (v) identification of the linkage between specific transcriptional reactions and the biological functions of these phytohormones [17]. The research on a model plant, *Arabidopsis thaliana* investigating indirect organogenesis, was a source of the great majority of these findings, although nowadays some comparative analyses of engaged sequences of other species are available (*Populus tremula*, *Oryza sativa*, *Zea mays*) [18,19,20,21,22]. The influence of cytokinin introduction on gene expression has been known for more than three decades, but it was the affordability of large-scale transcriptome analyses that initiated identification of many of such genes, revealing their up or downregulation. However, in most cases, there is still no functional connection to the inducing phytohormone.

### 3.1. Cytokinin Receptors

In *Arabidopsis* plants, cytokinin signaling starts with their detection by a set of three sensor receptors of a catalytic type. Discovered in 2001, the receptors, known as Arabidopsis Histidine Kinase 4 (AHK 4 or Cytokinin Response 1, CRE 1), Arabidopsis Histidine Kinase 2 and 3, are transmembrane proteins localized in the endoplasmic reticulum (predominantly) and the plasma membrane [23,24]. These CK receptors were initially studied in heterologous microbial systems (mainly bacteria), and later, to avoid protein properties distortion caused by the unnatural environment, and to provide a more accurate picture of their protein interaction, they were investigated as a part of the plant membrane [25]. They display domain structural similarity accompanied by a similar molecular weight of ca. 100 kDa. These multidomain receptors consist of a sensor Cyclase/Histidine kinase Associated Sensory Extracellular (CHASE) domain, displaying cytokinin binding activity, as well as at least two sensor flanking transmembrane domains: a catalytic sequence displaying histidine kinase activity comprising an A domain, which allows dimerization, an ATP/ADP binding phosphotransfer domain and a receiver domain located at the C-terminus [26].

Cytokinins, the target molecules of these receptors, occur in plant organisms as a variety of isoforms. The most prevalent among those is zeatin, which can be found in *trans*- and *cis*- configurations, followed by N9-riboside phosphate and the N9-ribosylated derivatives, while isopentenyladenine and dihydrozeatin are also presented at a reasonable, but lower, level. The aromatic types of phytohormones (N6-benzyladenine, topolin) are present in plant tissues as well but at smaller concentrations [27]. The tZ riboside and iP riboside are considered to be the major transport isoforms of CKs, translocated through xylem and phloem structures. Interestingly, according to some studies, it is actually the concentration of cytokinin ribosides that could be interpreted as an indicator of the availability of active CKs. According to the other researchers, it is still questionable if the ribosides are the active forms of cytokinins [25]. To a certain extent, this also influences the process of cytokinin signaling, because it remains unclear whether, for example, *trans*-zeatin riboside, once transported via the xylem to other parts of a plant, requires conversion to the corresponding free base form to trigger shoot organogenesis, or whether it happens as a consequence of its direct binding to the receptor. A similar inconsistency was also observed regarding the interaction of hormone receptors with cytokinin ribosides. However, the observation of the crystal structure of the AHK4-tZ complex should shed light on the biological activity of ribosides. The analysis of the AHK4-tZ complex revealed that the positions of the N3, N9, and N7 of the adenine ring are buried in the binding pocket of AHK4. This observation supported the absence of the hormonal activity of tZ ribosylated at the N9 position because the riboside moiety did not fit into the binding pocket [28]. Interestingly, despite the above, earlier studies conducted on a membrane system from *Escherichia coli* favor riboside ability to bind the receptor. However, investigating the competition of isopentenyl adenosine, *trans*-zeatin riboside and their corresponding bases, Lomin et al. [25] showed that the efficiency of the creation of the hormone-receptor complex depends greatly on the membrane system involved. In a system of *E. coli* spheroplasts, ribosides were as effective as the free bases. However, when plant membranes from the leaves of *N. benthamiana* were used, they hardly competed for binding to the cytokinin receptors [25]. Hence, to obtain results that would allow for a reliable conclusion concerning this issue, plant membrane assays seem to be necessary. Even so, compelling evidence indicates that any such extrapolation of such results to other plant species should be done with caution, perhaps involving a phylogenetic back-up.

Various analyses confirmed that the cytokinin receptors complement each other, regarding both their functionality and localized expression [27]. Indeed, a mutation in a single receptor seems to have no observable effects on plant phenotype, while double or triple inactivation results in serious consequences [29]. In like manner, the localization of the cytokinin receptor is certainly not accidental. The AHK3 receptor is predominantly found in leaves, while AHK4 occurs mainly in roots, which seems to be consistent with the ligand specificity of the receptors and the outcome of transport processes [27,30,31]. It has been shown that CK receptors display a differentiated ability to perceive different cytokinins and their derivatives. The aforementioned leaf sensor AHK3, an orthologous counterpart of ZmHK2 from *Zea mays*, reveals the highest affinity for *trans*-zeatin and decreasing affinity for dihydrozeatin >isopentenyladenine >N6-benzyladenine >*cis*-zeatin (see structures; Figure 1). It seems naturally connected with the influx of its dominant target cytokinin from roots [25]. In turn, the AHK4 (CRE 1) and its corresponding ZmHK1 receptor are characteristic of roots, and predominantly detect isopentenyladenine, the main representative of cytokinins in the phloem. However, it does not mean that AHK4 is not sensitive to *trans*-zeatin; it just displays lower affinity for this type of CK [22,25].

In general, however, the question of affinity should be carefully explored. For example, although the *Zea mays* receptors above demonstrate similar ligand specificity to their counterparts from *A. thaliana*, care should be taken when generalizing such affinity findings to other species, while *cis*-zeatin, a weakly-active, or inactive isoprenoid, reacts with *ZmHK1*, *ZmHK2* and *ZmHK3a* receptors, it demonstrates very little reactivity with *Arabidopsis* histidine kinases(AHKs) [22]. Therefore, it seems that the side chain of isoforms has a considerable influence on receptor binding.

### 3.2. Molecular Background of CK Signal Transduction

The cascade of protein phosphorylation is a chief mechanism regulating routes of signal transduction in prokaryotic and eukaryotic cells. The cytokinin signaling is structurally and functionally related to the bacterial Two-Component System (TCS), which is believed to be one of the most common systems in prokaryotic organisms [26,32]. It has been extensively explored and quite well-understood [33]. Typically, the canonical bacterial TCS consists of two elements. These are a sensor histidine kinase that is activated and autophosphorylated under the presence of external factors, and a single response regulator receiving the high-energy phosphate. Such systems have also been defined in numerous eukaryotic organisms, except for animals. They were reported in 1993 in plants and fungi [34]. Usually, such a phosphorelay cascade includes three subfamilies of involved molecules creating a system termed Multistep-Phosphorelays (MSP) (also occurring in prokaryotes). The insights into both the multistep-phosphorelay and two-component system have been presented in a detailed and accessible way in the paper by Mira-Rodado [33].

The model thale cress represents the most investigated plant. There are, for example, 16 *Arabidopsis* histidine kinases known that could be classified as hybrids because of the presence of histidine and a receiver domain in one molecule. According to various authors, the *Arabidopsis* MSP system includes 11 HKs [33]. They typically contain at least His or Asp conserved residues (mostly both) in their structure. Interestingly, none of the phosphotransfer proteins or response elements of this species is responsible for a completely independent pathway, not being involved in cytokinin signaling [34,35]. This fact shows how multitasking and important the cytokinin hormones are for plants. However, although cytokinins influence plant fate via a phosphorelay cascade, compelling evidence proves that the system itself is evolutionarily older. The system, and the histidine kinases, or type-B response regulators in particular, have been found in unicellular organisms. It seems interesting that such a complex signal transduction cascade, which can direct a cell specialization and multilevel crosstalk in multicellular organisms, has been developed in single-cell prokaryotes. In plant MSPs, a third element, the His-containing phosphotransfer protein, has been included and kept in a relatively unchanged form. It may seem that MSP is a complex outcome of evolution while the nature of the phosphorelay remains constant [33,36]. In angiosperms, some system variation is observed within its structure, and new types of CK receptors based on serine/threonine phosphorylation are discovered. They probably arose while dicots and monocots diverged from each other [37]. In spite of this, data obtained from research on different species (*Medicago sativa*, *Glycine max*, *Oryza sativa*, *Zea mays* or *Solanum lycopersicum*) show that phosphorelay elements are quite conserved across the plant world [35].

Described in *Arabidopsis* plants, cytokinin signaling engages three modules: cytokinin-binding histidine kinase receptors, a set of phosphorelay shuttle proteins responsible for transduction along the pathway cytoplasm—nucleus structures and type-B Response Regulators (RRB) that are transcription factors. The last elements of this cascade activate gene expression related to cytokinin primary responses. Additionally, a negative feedback on multistep-phosphorelay signaling is exerted by type-A response regulators [32]. Apart from conventional classification into type-A and type-B, type-C response regulators have been defined but their role remains elusive [37,38]. In general, the signal has to pass two membrane systems: first, the plasma membrane/ER membranes (depending on exo or endogenous origin of the phytohormone) and then, a nuclear membrane. In de novo shoot induction, the cytokinin molecule is perceived by a dedicated receptor, for example, in the case of root explants, the isopentenyladenine isoform is more likely to be detected by CRE 1/AHK4, although there is rather no strict correlation between a receptor type and shoot induction at this level [26,30]. In brief, upon cytokinin signal perception and binding by the CHASE domain, the autophosphorylation of a histidine (H) residue in a conserved site (H-box; **A**TV**SH**E**IR**TP) of the kinase domain is performed. Four conserved boxes, known as N-, G1-, F- and G2-motifs, are believed to participate in the binding of ATP molecules. Finally, the C-terminus receiver domain, including an acceptor sequence (DD-**D**-K) with the key aspartate residue, mediates the transport of the phosphate moiety [26], and thus, the first boundary membrane is breached.

The next step is assured by a set of low molecular weight mobile Histidine Phosphotransfer Proteins (HPT). These phosphotransmitters also possess the conserved histidine residue and shuttle between cytoplasm and the nucleus transferring the phosphate signal. In thale cress, two classes of these MSP have been distinguished. One of them relates to HPTs and carries the aforementioned H residue being able to pass the signal from receptor domains towards nuclear type-B response regulators (positive regulators). The second group possesses asparagine instead of histidine and, therefore, cannot run the phosphotransfer (negative regulator) [32]. The cytoplasmic phosphotransmitters transfer the phosphate moiety to the nucleus and there activate the type-B response regulators. These regulators act as transcription factors containing some typical structures; namely at least one Nuclear Localization Signal (NLS; probably in carboxy-terminus variable region), a GARP-like DNA-binding (sharing common features with MYB transcription activators) and transactivation domains. In addition to this category, there are two more aforementioned types of response regulators, A and C. In *A. thaliana* species, all known regulatory elements were divided into three aforementioned groups basing on their structure and function [33]. A conserved aspartate (Asp) residue was detected in the receiver domains of all categories of RR, but only the type-B was considered to have TF function [39,40]. When phosphorylated, the final components of MSP, the B-type RRs regulate the expression pattern of different cytokinin-related genes, including, among others, the A-type RRs genes assuring a dampening effect if a cytokinin signal is prolonged [38]. The B-RRs enter the nucleus and bind to target sequences. Subsequently, their transactivation domain stimulates the activation of downstream cytokinin-induced transcriptional responses. Interestingly, it is thought that, at a low cytokinin concentration, the receiver domain (RD) can veil the DNA binding module due to its conformation. In this manner, the upstream phosphorelay to response regulators is blocked until the higher level of cytokinin is detected and influences the spatial structure of RD [41]. It should be noted here that a parallel branch may mediate the phosphorylation process at this stage in planta. The phosphoryl group may be donated from HPT proteins to Cytokinin Response Factors (CRF). They show certain similarity to Type-B RRs, functioning as transcription factors as well. However, apart from sharing some target sequences, they also interact with their own ones [41].

The scheme of the cytokinin signal transduction is presented below (Figure 2).

### 3.3. Regulation of Cytokinin-Related Genes in Shoot Development

De novo shoot formation requires initiation and maintenance of the functional shoot apical meristem (SAM) with its central zone (CZ) carrying the pool of pluripotent stem cells (SC). The central zone includes three separate cell layers that form the epidermis (L1), ground tissue (L2) and vasculature (L3). The induction and maintenance of the stem cell fate are conditioned by the activity of the organizing center (OC), which is situated directly underneath the SC area [42,43]. The transcriptional activation of *WUSCHEL* (*WUS*) is considered to be a critical molecular event triggering CK-induced shoot organogenesis. In the process of SAM establishment, *WUS* is thought to define the organizing center. Therefore, *WUS* expression is substantial for the de novo organization of the shoot stem cell niches that generate signals that control the balance between self-renewal processes and the production of daughter cells that can differentiate into new tissues [43], and induction of stem cell fate in the pool of cells that overlie the OC. Meristem maintenance depends upon a balance between stem cell division within the meristem center and differentiation toward the periphery. Ectopic shoot formation is clearly visible as a result of *WUS* overexpression. Since *WUS* mutants displayed no shoot regeneration in vitro, it is assumed that stimulation of the identity acquisition of the shoot meristematic progenitor cells requires a check point level of *WUS* gene expression [44]. Clearly, the cells designated as *WUS*-positive mark the shoot progenitor region during de novo shoot organogenesis [45]. *WUS*, a homeodomain transcription factor, is synthesized in the organizing center (its RNA is present in cells of the OC niche) and then moved to a central zone to activate *CLAVATA3* (*CLV3*), a specific regulator of shoot meristem development. The *WUSCHEL*–*CLAVATA3* expressing system makes a core regulatory loop coordinating proliferation and stem cell identity in the shoot apical meristem (keeping a constant number of stem cells) [43]. Once *WUS* migrates to the CZ, it binds to a promoter sequence of *CLV3* (using three WUS-binding elements with conserved TAAT motif). The latter gene encodes a small peptide (a negative feedback regulator) that binds to CLAVATA1 (CLV1) in the extracellular space. CLV1 is a leucine-rich repeat receptor kinase typically synthesized in the rib meristem (RM) area. Its activation (along with other receptor kinases) mediates the downregulation of *WUSCHEL* transcription in OC cells [46]. Additionally, *WUS* TF directly suppresses the transcription of *Arabidopsis RR7* and *RR15* genes in the cells of the organizing center. Hence, it indirectly promotes cellular cytokinin response by blocking its cellular A-type inhibitors [42,47]. Apart from *WUSCHEL* or *CLAVAT3*, the stage of shoot induction is connected with the transcriptional activation of other shoot meristem-associated genes, for example, the *SHOOT MERISTEMLESS* (*STM*) gene, which is expressed in the promeristem [46]. It is considered to be a key switch responsible for meristem maintenance, but it is apparently a minor factor in the de novo setting up of SAM [45]. It keeps SAM cells in an undifferentiated form independently of WUSCHEL and is necessary to promote cell division. Some other examples of the genes involved in shoot organogenesis are gathered and presented in Table 1. In-depth analysis of its genetic background is provided, e.g., in papers by Shin et al. [15], Zubo and Schaller [48] or Ikeuchi et al. [3]. Interestingly, plant shoot organogenesis, based on a guideline provided by phytohormones (here, cytokinins), does not invariably follow the exact pattern of embryonic development. Just recently, the studies by Zhang et al. [45] indicated that embryogenesis and de novo shoot establishment might engage different genetic programs to form the SAM structure, e.g., the lack of *WOX2* (expressed at the zygote level) transcripts in wild-type explants on shoot inducing medium [45].

On a side note, most subject reviews focus on indirect organogenesis, where the development of the apical shoot meristem follows auxin addition, which results in callus formation and engages auxin-induced genes [15]. The classic pathway of de novo shoot induction, involving tissue culture on CIM and SIM, seems to be more popular and easier to achieve for many plants, although indirect, auxin-engaging mechanisms may have a more complex genetic background. Genes like *YUC1*, *YUC4*, *PLETHORA* (*PLT3*, *PLT5*, *PLT7*) or *CUP-SHAPED COTYLEDON* (*CUC1* and *CUC2*) may be earlier attached [3,15]. Moreover, a great number of in vitro protocols assume the use of auxin (along with higher cytokinin concentrations) just to trigger processes of obtained bud/shoot elongation, without a callus intermediate. The supplementation with small amounts of auxin can stimulate the bud/shoot growth instead of callogenesis (if any, it develops only at the bases of explants); especially, IAA is used for the elongation effect, since it is much less callogenic in comparison to 2,4-D and NAA. However, resulting from such indirect or auxin-based protocols, hormonal crosstalk is distinct from the one observed when de novo shoots are generated as a direct response to a cytokinin-rich environment. Additionally, quite a large number of species is able to develop SAM without any auxin stimulation. It seems that, in these cases, genetic networks of auxin and cytokinin may converge mostly at the level of cytokinin-dependent repression of auxin-related genes. The availability of exogenous cytokinins does not require strong induction of genes responsible for their biosynthetic enzymes (like *ISOPENTENYLTRANSFERASE 3* or the *LONELY GUY* gene family: *LOG1*, *LOG4*, *LOG5* in the case of wounding event), although, on the other hand, the CK biosynthetic gene *LOG2* is upregulated in response to cytokinin. On the one hand, it potentially serves to help replenish the cytokinin pool, and probably to achieve the cytokinin amount needed to trigger the repression of their excessive biosynthesis on the other [3,48,71]. Of course, it seems obvious that the direct process should also engage the genetic network connected with the induction of pluripotency in the explant cells. Nevertheless, the intention of this review was to indicate genes connected with the phenomenon of pure cytokinin-related shoot direct organogenesis. However, it turned out to be only an attempt at a fragmentary look at a powerful, complex hormonal and genetic grid, impossible in a way.

The addition of cytokinin causes rapid down and upregulation of many genes (e.g., via influencing their steady-state mRNA levels). In fact, it seems that the CK-mediated de novo shoot development is the consequence of: (i) the expression of genes involved in cytokinin balance, (ii) signal transduction, (iii) activity of factors responsible for the regulation of transcription process, (iv) splicing performing and (v) chromatin remodelers’ activity [72]. The shift of the root transcriptomes towards shoot formation depending on the presence and dose of the cytokinin, suggests that the phytohormone determines at least part of the organ-specific transcriptome templet, independently of its morphological identity. The B-type response regulators are the final module of a system that allows the cell to convert an external stimulus—a cytokinin—into a definite internal recommendation signal—a precise molecular/genetic instruction on further activity to be taken. It is at this level that one can look for a clearer connection between cytokinins and shoot organogenesis. Due to the miscellaneous research and combined efforts of many scientists, an inventory of potential binding motifs for Type-B response regulators has been compiled [48,73,74,75]. The cytokinin response genes were ascribed to the above-mentioned group on the basis of, among other things. The microarray expression analysis, RNA-seq data, chromatin immunoprecipitation-sequencing (ChIP-seq) studies and research on overexpressing or loss-of-function mutants [48,73,74,75]. In many cases, the sequence of DNA-binding motifs of different B-type response regulators overlap. Xie and co-workers [75] determined the DNA binding profiles investigating three regulatory elements of cytokinin response: ARR1 (At3g16857), ARR10 (At4g31920) and ARR12 (At2g25180). While analyzing the cytokinin network based on B-ARRs targets, it was shown that transcription factors shared different genes alternately (presented in the paper by Xie et al. [75]). As a conclusion, they identified certain similarity among B-RR *cis*-motif candidates, flanked by degenerate sequences [75]. The core nucleotide pattern consists of 5′-AGATHY-3′ (wherein A reveals slight degeneration towards G), where H stands for A, T, or C and Y for T or C residue. Promoters of genes that are upregulated by cytokinins are enriched with this pattern [48,75]. However, according to some authors, a clear ”sequence-motif-based classification of a cytokinin-inducible promoter is still missing”, and the lack of cytokinin response motif does not exclude promoter reactivity to cytokinins) [76]. CKs endorse B-ARR motif switching from a more degenerated sequence to a canonical AGATHY motif [75]. The impact of the cytokinin hormone via the transcription pathway in planta was initially demonstrated on nonsymbiotic hemoglobin (*NSHB*) genes. The transient bombardment test performed on tobacco leaf disks, using a rice OsNSHB2 promoter, revealed the cytokinin-dependent promoter activation. This activation was mediated by ARR1 via its binding to *cis*-element AGATT in the promoter sequence [17,52]. The outcome is consistent with the observation that ARR1 displays the uppermost CK-dependent intensification of binding to its genes. Without phytohormone treatment, ARR1 recognized (and, according to other hypotheses, loosely binds only) 2815 possible targets. Subsequent cytokinin treatment increased this number to 5128 (after 4 h), and then to 10,340 (when supplemented for three days) [75]. Summarizing, B-type RRs function at the top of a transcriptional cascade, and prevalent CK-regulated genes connected with de novo shoot regeneration are those that encode the secondary wave of TFs responsible for the feedback affecting the cytokinin response. According to dosage experiments, there may be even 10,000 cytokinin general response genes accounted in the B-RR network. Taking into consideration the fact that their activation might be limited mostly by the concentration/availability of the endogenous cytokinins that modify type-B response regulators, this may be still a challenging number.

As it has already been discussed, *WUS* is a strategic gene indispensable to shoot progenitor formation, shoot apical meristem maintenance (as a nonautonomous signal) and shoot morphogenesis. Referring to a cytokinin-dependent de novo shoot organogenesis, it was proved that B-type RRs, the fundamental players in cytokinin signaling, were primary positive regulators of the *WUSCHEL* expression [46]. Out of at least 11 B-RR family members in thale cress, the three aforementioned elements (ARR1, ARR10 and ARR12) are the most significant as far as the regulation of the majority of phytohormone-induced genes is concerned (recently, ARR2 has been listed as one of the master regulators as well). Examination of their double or triple mutants revealed limited explant reactiveness toward CKs and reduced competence to regenerate shoots [48], also due to a lack of *WUSCHEL* induction. Thus, these B-ARRs, apart from the activation of *WUS* transcription, take part in SAM maintenance, formation of axillary shoot meristems and in vitro regeneration of shoots. In parallel, they maintain a mutual phytohormone balance by repressing auxin accumulation by blocking the expression of YUCCAs encoding critical enzymes for auxin biosynthesis [53,75]. A thorough analysis exhibited the synergistic impact of different B-type RRs on processes of shoot organogenesis. Namely, *Arabidopsis* ARR1 and ARR12 significantly hinder root elongation; all three acting to maintain the size of SAM [62]. In *Arabidopsis thaliana* ARR1, ARR2, ARR10 and ARR12 directly induce the *WUS* factor gene and physically interact with partner microRNA165/6 targeted HD-ZIP III transcription factors: PHABULOSA, PHAVOLUTA and REVOLUTA (PHB, PHV and REV, respectively), to spatially restrict *WUS* expression to shoot progenitor cells. The inactivation of all these complex compounds gives a triple mutant (*phb*, *phv* and *rev*) with impaired ability to create shoot meristems. This loss-of-function event cannot be reversed even by a high dose of cytokinin [62]. Additionally, they upregulate other core factors for SAM formation, like transcription factor STM [3]. Strikingly, the analysis of the B-type relation network led to the discovery of class III HD-Zip proteins, which turned out to be pivotal developmental regulators in shoot formation. Another vital connection in the B-type grid of regulation targets, aside from the primary CK response genes or A-type response regulators, is the AHK4 gene encoding the shoot specific cytokinin receptor. Studies on thale cress revealed that the addition of zeatin resulted in considerable B-type element-mediated up-regulation of AHK4 protein levels. The B-RRs support cytokinin signaling directly inducing its expression within potential shoot promeristem and, importantly, *WUS* SAM-inducing activity, is confined to the cells marked by AHK4 receptor presence [42,75]. Thus, unquestionably, B-RRs are the key molecules that manage the transcriptional regulation of cytokinin signal transduction but, importantly, they are not regulated at the transcriptional level by CK but are the subject of post-transcriptional modification. However, certain reciprocal feedback is implicated, since several of their representatives (e.g., ARR1, ARR10, ARR12, ARR14, ARR18) were discovered to be among target gene candidates actually transcriptionally controlled by B-type RRs [75].

As it has already been mentioned, the B-type ARR1, ARR10 and ARR12 directly activate the homeobox *WUS* TF. Scientific research indicates that B-type ARR12 is one of the most powerful positive regulators of shoot regeneration. It seems unquestionable that it does not contribute to dedifferentiation or callus proliferation processes but is engaged in the formation of shoot apical meristem. The loss-of-function mutant hardly regenerates shoots, while *ARR12* overexpression clearly amplifies the number of the differentiated shoots (on a side note, the *arr12* mutant displays up-regulation of the aforementioned intracellular A-type inhibitors, *ARR7* and *ARR15*; thereby, the shoot organogenesis is hindered twice) [44,47,77]. ARR12 acts as a linkage between cytokinin signal transduction and the specification of shoot apical identity during shoot regeneration. The chromatin immunoprecipitation revealed the target genes of this regulator and the connector point. B-type transcription factor binds straight to multiple copies of (A/G)GAT(T/C) cytokinin motif in the *WUS* promoter, triggering the synthesis of the WUSCHEL protein [44].

As with all B-type elements, ARR1 was usually described as a positive regulator and drew much attention [52]. The aforementioned example of the activation of rice OsNSHB2 promoter by ARR1 is further evidence of the importance of these regulatory elements in shoot regeneration. It was demonstrated to bind, in particular, to the extended cytokinin response motif, ECRM, which comprises the octameric sequence AAGAT[T,C]TT [74,76]. This factor functions locally during shoot organogenesis and it was shown to affect the transcription of primary CK response genes. At least 17 sequences encoding proteins of different functions are upregulated by ARR1 (including A-type RRs, enzymes of cytokinin metabolism and putative stress resistance proteins) [53,74]. Surprisingly, just recently, it has been reported that ARR1 acts as an important inhibitor of in vitro shoot organogenesis as well. According to Liu et al. [54], the phenomenon of the repressive activity of B-type ARRs towards genes involved in the shoot organogenesis has not been previously described. Independently from cytokinin dosage, root and hypocotyl explants of the *arr1* mutant produced more shoots than the wild type (WT) plants. ARR1-related repression of callus induction and shoot regeneration is connected with the presence of the ARR12 factor. It has been hypothesized that it moderates the expression of *WUS* and *CLV3* genes in ARR12-dependent mode and that it directly initiates *INDOLE-3-ACETIC ACID INDUCIBLE17* (*IAA17*, a repressor of auxin response gene). The fact that *CLV3* expression is repressed by ARR1 suggests that it functions upstream of at least *CLAVATA3*. The two B-type regulators may affect *WUS* expression by competitively binding to the same motifs in the *WUSCHEL* promoter. Apparently, ARR1 counteracts the positive impact of B-ARR12 provoking inhibition of callus creation and shoot development. Thereby, ARR1 may block the regeneration of shoots. The data collected by Liu and co-workers suggest that overexpression of ARR1 slightly enhances shoot induction in the *arr1*, *arr12* double mutant (while CK signaling is blocked). On the other hand, the WT plants with ARR1 overexpression displayed shoot regeneration inhibition. The function of ARR12 might be altered then. Thus, a B-type network comprises response regulators of various and putatively shifting strength (where, in *Arabidopsis**,* ARR12 is a key positive *WUS* regulator and ARR1 is a weak positive one) [54].

The activity of ARR10 seems to raise fewer doubts. Similar to the two above-mentioned elements, it functions locally during shoot organogenesis and directly binds to the *WUSCHEL* sequence [53]. According to the authors’ best knowledge, the inhibitory effect on shoot organogenesis has not been proved in the case of ARR10. In 2017, Zubo et al. [78] demonstrated its cytokinin-dependent enhancing influence on *WUS* expression [78]. The analysis of CK hypersensitive lines of thale cress (derived from the ARR10 ectopic overexpression) allowed clarification of the involvement of cytokinins in the modulation of physiological responses in planta. This pointed out a number of potential target candidates that may be responsive to this transcription factor (with prevalent DNA regions associated with the A-type ARRs). It also turned out that ARR10 shared binding motifs with ARR1 and with other B-type regulators like ARR11 or ARR14 [78,79]. Referring to the latest findings by Liu and coworkers [54], the counteracting relation of ARR1 with all these factors (and with ARR10 in particular) cannot be excluded.

#### Epigenetic Reprogramming in Cytokinin Signaling

Apart from all the above factors, genetic reprogramming conditioning de novo shoot organogenesis is also the function of chromatin availability. Thus, epigenetic mechanisms represent another level of controlling plant cell regenerative capacity. Since loci of regeneration-related target genes can be epigenetically repressed, there must also be a state of permissive chromatin, leading to the transcriptional activation of shoot meristem formation. Jerzmanowski [80] distinguished two major groups of agents that regulate chromatin dynamics. Namely, (i) remodelers of chromatin, which switch interactions between DNA and histones using energy from ATP hydrolysis and (ii) enzymatic nucleosome-modifiers that specifically introduce or remove covalent modifications and, in this way, modulate DNA and histone residues [80]. The regulators unnecessary for the maintenance of SAM, such as WUSCHEL and SHOOT MERISTEMLESS, are subject to epigenetic control. The expression of *WUSCHEL* in *Arabidopsis* OC of the plant shoot apical meristem was shown to be controlled by epigenetic modifications (both, by methylation of DNA and modifications of histones) [81]. At the early steps of shoot regeneration, B-type ARRs are broadly expressed but *WUS* activation is not observed, which implies the involvement of other factors in *WUS* repression [45,53]. Indeed, several investigations have revealed that methylation of DNA, as well as modifications of histones (e.g., H3K9me2 and H3K27me3), repress *WUS* expression in the callus tissue [45,61,81].

DNA methylation in plants occurs at CG, CHG and CHH motifs (where H may stand for A, C or T). Compiling evidence shows that decreased levels of DNA methylation encourage de novo shoot formation in planta. The level of methylation is controlled, amongst other things by DNA METHYLOTRASFERASE 1 (MET1), an enzyme that may be modulated by many factors or states/events (namely, by E2FA; CYCD3; cytokinin-induced cell cycle) [61]. In model *Arabidopsis* plants, the loss-of-function *met1* mutant exhibited accelerated shoot regeneration on a shoot inducing medium with prematurely observed *WUSCHEL* expression [55]. MET1-mediated methylation of the *WUS* locus is considered to counteract cytokinin-promoted *WUSCHEL* activation. A more detailed description of MET1 regulation during shoot regeneration has been provided in a publication by Liu et al. [61].

Different epigenetic mechanisms also display certain crosstalk having common players. Chromatin remodelers responsible for nucleosome assembly/disassembly events, such as FASCIATA 1 (FAS1), may be directly regulated by the E2F transcription factor [82]. The latter controls one of the key events in the cell cycle transition: G1—S phase (the cell cycle repression brings limited capacity of shoot regeneration due to the lack of sufficient cells to form the meristem). The role of FAS1 and *FAS2* in SAM organization is confirmed by the studies on their mutants. Mutations in these genes lead to aberrant shoot apical meristems with an atypical expression of *WUS* [56,83]. Thus, on the one hand, E2F influences the CHROMATIN ASSEMBLY FACTOR 1 (CAF-1) complex via its subunits (FAS1, FAS2) (the complex is responsible for H3/H4 deposition) [82]. On the other hand, it controls the cell cycle, whose dynamics in cytokinin-rich milieu may affect the expression of METHYLTRASFERASE 1 [61]. High amounts of cytokinins may also endorse the removal of the repressive marks at the *WUSCHEL* locus in a cell cycle-determined way. Such histone marks (e.g., histone H3 lysine 27 trimethylation—H3K27me3) sustain the repressive status of different shoot regeneration-related genes (this type of methylation is conducted by a protein complex POLYCOMB REPRESSIVE COMPLEX 2, PRC2, which is evolutionarily conserved), also ensuring positive control of the shoot character and shape maintenance [3,62]. For example, the *STM* gene is expressed exclusively in the shoot apical meristem and repressed in lateral organ primordia. The expression pattern of *STM* is thought to be essential for SAM maintenance and its preservation is possible due to the repressive marker H3K27m3 in the *STM* promoter in primordia structure [56].

Among other epigenetic controllers, the SWI/SNF complex with the SPLAYED (SYD) and BRAHMA (BRM) factors (being ATP-dependent chromatin remodeling agents) may be found. These factors were demonstrated to be expressed in SAM and draw in shoot development. SPLAYED modulates *WUS* expression via direct binding to its promoter. Moreover, it limits *WUS* expression to the organizing center and thereby influences SAM organization and maintenance. Mutations in *SYD* abolish *WUS* expression and impair SAM formation [55,56].

The SWI/SNF complex is a subfamily of remodeling ATPases, originally purified from yeast, and it seems conserved in different organisms (fungi, plants and mammals). Members of this family are considered to be epigenetically engaged in plant stem cell initiation and maintenance. Just recently, it has been demonstrated that the SWI/SNF complex in *Arabidopsis* accelerates activation and repression, the expression of target genes binding to their *cis*-regulatory sequences (both promoters and terminators) [84]. Thus, chromatin availability clearly regulates shoot meristem assembly at different levels, e.g., by regulating the expression of various key players such as *WUS* or STM.

## 4. Conclusions

De novo shoot regeneration is one of the conditions allowing plant adaptation and survival in an unfavorable environment. It is also an essential prerequisite for plant propagation and genetic engineering. The plant life cycle is marked by the presence of stem cells that are capable of self-renewing and conversion into founder cells of a specific tissue. This ability underlies the process of shoot organogenesis and is unquestionably linked with hormonal guidelines. Cytokinins, phytohormones having pleiotropic effects, play a pivotal role in DNSO, which is clearly visible, and their signal transduction becomes more and more elucidated. Using the multistep-phosphorelay, CKs transduce information through membrane-localized histidine kinase receptors to finally activate nuclear B-type response regulators. These transcription factors trigger the cascade of genetic events, regulating both gene players that directly take part in metabolic response and those controlling the second line of transcription factors. This results in an impressive network of connections that crosstalk with multiple different life processes and are regulated at multiple levels (including transcription or epigenesis).

The aim of this review was to collect these aspects of cytokinin primary response that are responsible for de novo shoot organogenesis, but it turned out impossible in a way. The isolated analysis of genetic or molecular players responsible only for shoot organogenesis seems rather unlikely to succeed. Nevertheless, a certain fragmentary scheme of the cytokinin-dependent shoot induction has emerged. Most of the available analyses concern indirect organogenesis, explaining the role of callus intermediate and reciprocal auxin—cytokinin pathway connections. It results from the greater availability of indirect protocols. Perhaps more attention has been paid to this pathway resulting in more extensive knowledge, or it is the consequence of greater competence of most plant species to indirect regeneration, namely, it is easier to achieve in vitro. This may lead to the conclusion that, despite the complexity of indirect organogenesis, at a biochemical level it is easier for a plant to perform dedifferentiation to an unorganized state (callus) and then to redifferentiate to a target tissue, than simplifying to shift/reprogram directly one tissue type to another. Perhaps a direct process does not mean a less complex one.

Nevertheless, more thorough knowledge on players of cytokinin signaling mechanism in shoot regeneration may generate more questions about the process, and often requires a revision of the background theory. For example, the current studies clearly indicate that microRNAs are important factors of DNSO gene regulation, but satisfactory data on this subject are still missing from the literature [43]. The results by Liu et al. [54] shed light on the functioning of the signaling network, but challenge the previous findings on the interaction character of B-type RRs, which were considered to be positive regulators of DNSO [54]. These studies revealed a functional diversity among B-type regulators in modulating in vitro shoot regeneration (where RR12 acts as a key enhancer but RR1 turned out to be a substantial inhibitor of the process). Similarly, the research on the target motifs of transcriptional binding showed a significant need to explore the effect of phosphorylation on the selection of the target sequence. Moreover, most of the studies on cytokinin-dependent shoot formation were performed on the model plant *Arabidopsis thaliana.* Further research led to the identification of several putative orthologues of multistep-phosphorelays in other species (*Oryza sativa*, *Medicago trunculata*, *Glycine mays* or *Zea mays*). However, the products of the respective genes may not always have an equivalent function, and variations in elements of MSP have been described. Hence, as was mentioned, the extrapolation of molecules involved in cytokinin-dependent shoot regeneration should be careful. In addition, it seems obvious that the universality of the triggering cytokinin concerns mostly the primary response molecules. It should not matter what type of cytokinin is supplemented to the medium. However, there is no detailed research on the system variations that may occur due to the CK used, namely there is no, e.g., zeatin or benzylaminopurine-specific pathway described. The existence of differences can be inferred on the basis of plant morphology, growth rate or the presence of other markers. Many scientists reported variations in the regeneration of plant individuals (concerning morphology, secondary metabolites levels, etc.) after treatment with different cytokinins. It is very likely that they resulted from the activity of molecular players involved in the cytokinin secondary response. However, it requires further analysis to determine to what extent de novo shoot organogenesis and its underlying cytokinin is a universal response, even at the primary response stage.

## Figures and Tables

**Figure 1 genes-12-00265-f001:**
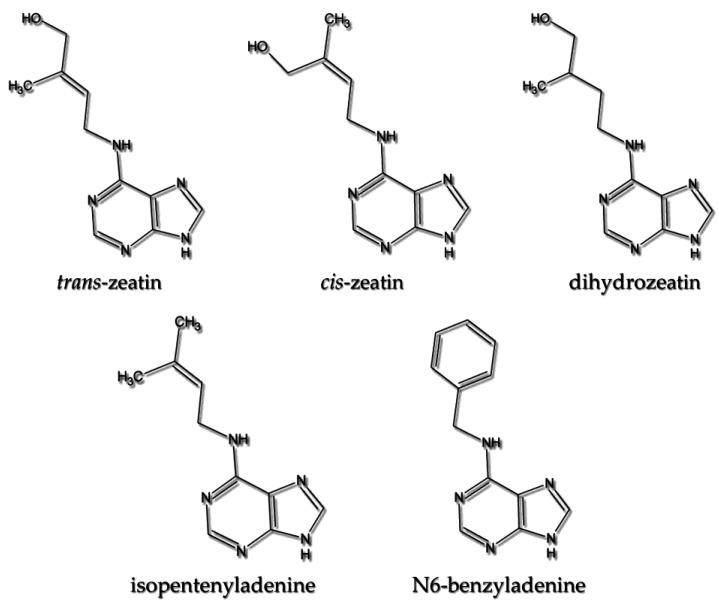
Structures of example purine cytokinins.

**Figure 2 genes-12-00265-f002:**
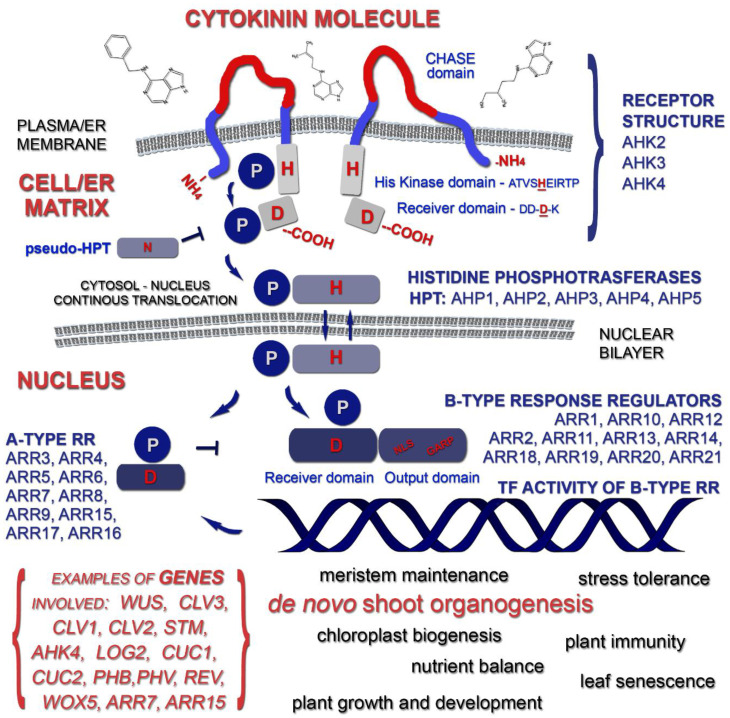
Cytokinin signal transduction in plants.

**Table 1 genes-12-00265-t001:** Genes involved in shoot organogenesis.

NAME ABBRV.	AGI LOCUS CODE	GENE DESCRIPTION	NOTES	Reference
AHK4/WOL	AT2G01830	Cytokinin signaling	ARABIDOPSIS HISTIDINE KINASE; ”shoot specific” cytokinin receptor	[23,24,49]
ARF3	AT2G33860	B3, ARF	TR; AUXIN RESPONSE FACTOR3; indirect blocking of cytokinin biosynthesis (represses *IPT5*—CK biosynthesis enzyme)	[3,50,51]
ARR1	AT3G16857	Type-B ARR	TR; regulator of *WUS* expression; stimulator/inhibitor (???)	[45,52,53,54]
ARR10	AT4G31920	Type-B ARR	TR; a positive regulator of *WUS* expression	[45,53]
ARR12	AT2G25180	Type-B ARR	TR a critical positive regulator of *WUS* expression	[44,45,53]
ARR7	AT1G19050	Type-A ARR	negative feedback on the multistep-phosphorelay signaling	[7,26,38]
ARR15	At1G74890	Type-A ARR	negative feedback on the multistep-phosphorelay signaling	[7,26,38]
BRM	AT2G46020	SWI2/SNF2 ATPase	ER; involved in ATP-dependent chromatin remodeling	[55,56]
CLF	AT2G23380	PRC2 subunit	ER; subunit of Polycomb Repressive Complex (2)—chromatin remodeling factors	[3,45,57]
CUC1	AT3G15170	NAC domain	TR; shoot-promoting TF; SAM initiation and cotyledon boundary establishment; promote *STM* expression	[3,45,58,59]
CUC2	AT5G53950	NAC domain	TR; shoot-promoting TF; SAM initiation and cotyledon boundary establishment; promote *STM* expression	[3,45,58,59,60]
CYCD3;1	AT4G34160	CYCD3 D-type cyclin	G1/S transition of mitotic cell cycle; regulation of cell population proliferation, expressed in SAM	[61,62]
CYCD3;2	AT5G67260	CYCD3 D-type cyclin	G1/S transition of mitotic cell cycle; regulation of cell population proliferation, expressed in SAM	[61,62]
CYCD3;3	AT3G50070	CYCD3 D-type cyclin	G1/S transition of mitotic cell cycle; regulation of cell population proliferation, expressed in SAM	[61,62]
E2Fa	AT2G36010	E2F	TR; positive regulation of cell cycle; influences chromatin remodeling	[3,61]
ESR1	AT1G12980	AP2/ERF	TR; TR, cytokinin response; DNA-binding TF	[63]
ESR2	AT1G24590	AP2/ERF	TR; response to auxin; cycle cell regulation	[64]
HAG1/GCN5	AT3G54610	GNAT/MYST	ER; histone acetyltransferase conducting histone modification	[65]
IPT3	AT3G63110	Cytokinin synthesis	adenosine phosphate isopentenyltransferase 3, cytokinin synthase, expressed in shoot apex	[50,55]
IPT5	AT5G19040	Cytokinin synthesis	adenosine phosphate isopentenyltransferase 5, cytokinin synthase	[50,62]
LBD16	AT2G42430	LOB	TR, involved in hormone-mediated signaling pathway; pluripotency acquisition	[59]
MET1/DDM2	AT5G49160	DNA methylation	ER epigenetic regulation of *WUS* expression	[61]
PHB	AT2G34710	HD ZIP III	TR, spatial developmental regulators in shoot formation, confining of *WUS* expression to shoot progenitor; STM upregulation	[3,45,62]
PHV	AT1G30490	HD ZIP III	TR, spatial developmental regulators in shoot formation, confining of *WUS* expression to shoot progenitor; STM upregulation	[3,45,62]
PIN1	AT1G73590	Auxin transporter	engaged in shoot and root development; callus formation	[62,66]
PLT3	AT5G10510	AP2/ERF	TR, AP2-domain TF; involved in formation of callus	[60,62,67]
PLT5	AT5G57390	AP2/ERF	TR, indirect influence on *WUS*-induced cell fate reprogramming and callus formation	[60,67]
PLT7	AT5G65510	AP2/ERF	TR, indirect influence on *WUS*-induced cell fate reprogramming and callus formation	[60,67]
RAP2.6L	AT5G13330	AP2/ERF	TR, involved in shoot stem cell specification	[68]
REV	AT5G60690	HD ZIP III	TR, spatial developmental regulators in shoot formation, confining of *WUS* expression to shoot progenitor; STM upregulation	[3,45,62]
SCR	AT3G54220	GRAS	TR, engaged in shoot and root development; callus formation	[62,65]
STM	AT1G62360	KNOX	TR, a key switch responsible for meristem maintenance; expressed in promeristem	[45]
SWN	AT4G02020	PRC2 subunit	ER, involved in ATP-dependent chromatin remodeling	[45,57]
SYD	AT2G28290	SWI2/SNF2 ATPase	involved in ATP-dependent chromatin remodeling, positive regulator of *WUS* expression	[55,56]
WIND1	AT1G78080	AP2/ERF	TR; reprogramming regulator, key role in formation of callus	[63,69]
WOX5	AT3G11260	Homeobox	TR; a member of the *WUS* family of homeodomain TF, acquisition of competency for shoot regeneration	[3,63,70]
WOX7	AT5G05770	Homeobox	TR; a member of the *WUS* family of homeodomain TF; acquisition of competency for shoot regeneration	[3,65,70]
WOX11	AT3G03660	Homeobox	TR; a member of the *WUS* family of homeodomain TF; acquisition of competency for shoot regeneration	[3,59,70]
WOX14	AT1G20700	Homeobox	TR; a member of the *WUS* family of homeodomain TF; acquisition of competency for shoot regeneration	[3,65]
WUS	AT2G17950	Homeobox	TR; strategic gene indispensable to shoot progenitor formation; *WUS* defines the organizing center in SAM	[45,53,66]
YUC1	AT4G32540	Auxin synthesis	OTHERS; YUC-mediated auxin biosynthesis is required for efficient shoot regeneration (callus)	[50,62]
YUC4	AT5G11320	Auxin synthesis	OTHERS; YUC-mediated auxin biosynthesis is required for efficient shoot regeneration (callus)	[50,62]

* TR—transcriptional regulation. * ER—epigenetic regulation. OTHERS—different type of regulation.

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
