# Peer review of "Cytokinin Signaling and De Novo Shoot Organogenesis"

_genes, 2021, doi:10.3390/genes12020265_

Round 1

Reviewer 1 Report

The manuscript entitled “The cytokinin signalling and de novo shoot organogenesis” authored by Hnatuszko-Konka et al. reviews the regeneration mechanism in plants, specifically focusing on the shoot organogenesis and the involvement of the cytokinin phytohormones in this process.  

Firstly, the authors briefly explain the distinctions of two types of regeneration in plants, organogenesis and somatic embryogenesis , then they dissect the overall cytokinin signal transduction pathway (receptors, phosphorelay cascade, transcriptional response) and the specific cytokinin-related response during shoot organogenesis, lastly they analyse the epigenetic control of the de novo  shoot organogenesis.

I appreciated the choice of the topic, which is timely and of great interest for a broad readership - not only for that with plant-centred interests. I also appreciated that the concepts presented in the manuscript are easily accessible, particularly for students, and that the authors were consistent with the aims they claimed in the introduction.

I noticed that, due to the vastity of the topics, particularly when speaking of cytokinin signalling in general,  the level of insight is not always deep, and the argumentations are sometimes a bit vague (for example, in line 244, how do the Type-A ARR negatively feedback on the cytokinin MPS?). For this reason, I think the authors could have addressed the cytokinin signalling not in general, but only in relation to the cytokinin transcriptional response involved in de novo shoot organogenesis.

Also, I perfectly understand that the authors are not English mother-tongue, but the vocabulary, the phrasing and the construction of many sentences throughout the manuscript sometimes sound a bit weird and make the text not always easy to read.

Summing up all my considerations, I think that the review needs some corrections and implementations, and that it can be made more informative and readable also, but not only, in the light of the following suggestions and comments:

  • Can the authors argument a little more in the introduction on the reason why epigenetic control is important during de novo organogenesis in relation to cytokinin in order to inform the reader at the beginning of the review? It becomes partly evident only later, in the corresponding section and in line 295. Including this topic because it takes part in de novo organogenesis is obviously correct, but it seems like a random choice among other processes that play a role.
  • The sentences in lines 193-196 need a more specific reference, like one or more research article(s) showing the expression patterns of the cytokinin receptors in planta. To my knowledge, AHK3 and AHK4 are not exclusively confined to the leaves and to the roots, respectively.
  • The sentence in line 196-197 absolutely requires a specific reference for Arabidopsis and not only for maize in the lines below.
  • In line 230-233 I would explicit the cause-effect node of the sentence to make it clearer for readers. Can you explicit why it is worth noting that the phosphorelay cascade is an evolutionary old mechanism when reasoning on its involvement in cell fate specification?
  • The sentence in brackets in lines 247-249 need a reference.
  • In line 267, the authors write:” In addition to this category, there are two more types of response regulators, A and C. “. But they already mentioned Type-A ARRs in line 244. Please, be consistent.
  • Sentence in line 292 requires a reference, please.
  • The sentence in lines 299-301 is not crystal-clear: Can the authors formulate it in a different manner?
  • Can the authors change the order of the information and data presented in the paragraph 3.3., called “Regulation of cytokinin-related genes in shoot development”? I mean: Can they re-build that paragraph, to make it internally coherent? It now begins analysing ARR-B, then it switches to WUS-CLV, then to auxin-cytokinin crosstalk, then it goes back to type-B cytokinin response regulators and WUS, to ARR-bs and HD-Zip III -dependent repression of WUS, then again it speaks about ARR-bs (12, 1 and 10) and WUS induction... It’s confusing and winding, and it represents the core of the whole review!
  •  in lines 364-365 the authors state that “in Different studies” it’s reported that de novo shhot organogenesis engages different genetic programs from that of embryogenesis, but then they report only one reference. Where are the other studies referenced?
  • What does the sentence in lines 389-91“although on the other CK biosynthetic gene LOG2 is up regulated in response to cytokinin (probably to help complement the phytohormone pool, perhaps to trigger the repression of CK biosynthesis)” mean? I think there is an error

  • I would add a stand-alone paragraph for epigenetic reprogramming because now it is part of the “regulation of cytokinin-related genes in shoot development” section, and even if it relates also to the arguments discussed there because cytokinins control methylation marks on the WUS promoter, this section broadens a lot the topic presented in the title and doesn’t fit totally with it.

Comments on the language:

  • Throughout the manuscript, the language register sounds often colloquial (for example in line 224 “platform” to refer to a model organism) and generalist, as in line 211, where the authors should be more specific. Which “Adequate system” are the authors referring to? In line 214, “It is the most common device in prokaryotic organisms [27, 30].”. Which common “device”? In lines 285-288, besides the fact that the sentence sounds weird and the punctuation is wrong, what is a “recommendation signal”? In line 345, “CLAVATA3 (CLV3), also a transcription factor” is incorrect, and the authors themselves specify in the following lines that it is a small peptide.
  • there is a general use of adjectives and adverbs that should be removed to make the text more fluid and less informal, for example in line 220 “cascade includes more (three) subfamilies” can be corrected with “cascade includes three subfamilies”, in line 397 “Undoubtedly” should be removed, and so on.
  • the use of brackets in the manuscript is wide but not optimal. some sentences within brackets should be removed as they sound like authors’ personal considerations without any bibliographical reference, as in line 174 and in line 384; some others bear important information/assumptions that should be put outside the brackets and discussed, as in lines 202-203, in line 226, in lines272-273, in lines 336-338 and also in lines 358-359; in some other cases a reference should be added, as in line 225, and then discussed

There are typos throughout the ms. I only report few exemplifications:

- in line154-155 “Arabidopsis Histidine Kinase 4 (AHK 4 or Cytokinin Response 1), Arabidopsis Histidine Kinase 2 and 3” all the name of the genes must be written in capital letters

- “nuclear” instead of “nucleus” in line 246

- In line 258, “between cytoplasm and a nucleus” should be corrected in “the cytoplasm and the nucleus”

- “mediate” instead of “intermediate” in line 279

- in the end of line 293, add “,”

- in line 343 write “WUS, a homeodomain transcription factor, is synthetized …” instead of “A WUS, homeodomain transcription factor is synthetized…”

- in line 350 “leucine-rich” instead of “leucine rich”

- in line 383, “cytokinin-rich” instead of “cytokinin rich”

- in line 440, “genes” instead of “gene”

-in line 580, add “about” between “studies” and “Cytokinin-“

Author Response

Responses to Reviewer #1

Dear Reviewer #1

I would like to thank you for the insightful revision. I appreciate all your comments and remarks.

Following your suggestion the manuscript has been edited by a native English speaker (certificate attached).

Regarding your comments:

  1. I noticed that, due to the vastity of the topics, particularly when speaking of cytokinin signalling in general, the level of insight is not always deep, and the argumentations are sometimes a bit vague (for example, in line 244, how do the Type-A ARR negatively feedback on the cytokinin MPS?). For this reason, I think the authors could have addressed the cytokinin signalling not in general, but only in relation to the cytokinin transcriptional response involved in de novo shoot organogenesis.

Can the authors argument a little more in the introduction on the reason why epigenetic control is important during de novo organogenesis in relation to cytokinin in order to inform the reader at the beginning of the review? It becomes partly evident only later, in the corresponding section and in line 295. Including this topic because it takes part in de novo organogenesis is obviously correct, but it seems like a random choice among other processes that play a role.

– we perfectly understand this remark. While preparing this manuscript we considered subjects that should/could be discussed, trying to present this phenomenon in a possibly holistic way. To stress its complexity we decided to cover some issues (also to keep the text comprehensible to scientists working outside the topic of the MS), even at the cost of a little cursory discussion. Perhaps this is a wrong assumption, but it is very difficult and somewhat artificial to isolate genes and transcriptional regulators related only to the organogenesis of shoots. It became very clear to us while reviewing this process. On the other hand, there are many excellent reviews dedicated to more detailed aspects of cytokinin MPS and its involvement in de novo shoot organogenesis. Therefore, we tried to present its complexity, to keep balance in this presentation and to build our individual approach to the subject. Sometimes we actually express our assumptions, as you noticed – mostly in brackets, to stress something or to open a discussion.

Referring to epigenetic control, we have added a brief explanation reasoning such a choice. We are aware that there are many levels of regulation that play a role, especially taking into a consideration impressive crosstalk of regulators engaged, but on the other hand, the activation of key transcriptional regulators, dynamic changes in gene expression and phytohormone interaction occurring in organogenesis are usually presented in the subject literature. The role of epigenetic control is rather seldom discussed, and its influence its sometimes underestimated (line 70).

Addressing your comments we have also tried to be consistent with opinions presented by two other reviewers, however we do hope we managed to explain all the issues.

  1. The sentences in lines 193-196 need a more specific reference, like one or more research article(s) showing the expression patterns of the cytokinin receptors in planta. To my knowledge, AHK3 and AHK4 are not exclusively confined to the leaves and to the roots, respectively.

– following your suggestion the required references have been added; perhaps we did not stress it clearly enough; these receptors are not exclusively confined to these organs, therefore we used words “mainly” or “predominantly” (line 206);

  1. The sentence in line 196-197 absolutely requires a specific reference for Arabidopsis and not only for maize in the lines below.

– the cited publication by Lomin et al. (2015) refers to studies on affinity of Arabidopsis receptors (line 213).

  1. In line 230-233 I would explicit the cause-effect node of the sentence to make it clearer for readers. Can you explicit why it is worth noting that the phosphorelay cascade is an evolutionary old mechanism when reasoning on its involvement in cell fate specification?

– it has been clarified. We found interesting, that the system, and the histidine kinases or type-B response regulators in particular had been found in unicellular organisms and that such a complex signal transduction cascade, which could direct a cell specialization and multilevel crosstalk in multicellular organisms, had been developed in single cell Prokaryotes (of course, primary dedicated to receive environmental stimuli). It may seem that MSP is a complex outcome of evolution while the nature of the phosphorelay remains relatively constant (lines 249 – 257 ).

  1. The sentence in brackets in lines 247-249 need a reference

– the required references have been added (line 277).

  1. In line 267, the authors write:” In addition to this category, there are two more types of response regulators, A and C“. But they already mentioned Type-A ARRs in line 244. Please, be consistent.

– it has been corrected (line 268);

  1. Sentence in line 292 requires a reference, please.

– the required reference has been added (line 398).

  1. The sentence in lines 299-301 is not crystal-clear: Can the authors formulate it in a different manner?

– the sentence has been modified (line 421).

  1. Can the authors change the order of the information and data presented in the paragraph 3.3., called “Regulation of cytokinin-related genes in shoot development”? I mean: Can they re-build that paragraph, to make it internally coherent? It now begins analysing ARR-B, then it switches to WUS-CLV, then to auxin-cytokinin crosstalk, then it goes back to type-B cytokinin response regulators and WUS, to ARR-bs and HD-Zip III -dependent repression of WUS, then again it speaks about ARR-bs (12, 1 and 10) and WUS induction... It’s confusing and winding, and it represents the core of the whole review!

– the paragraph has been slightly re-built. However, please note that some of the signaling players have been mentioned e.g. as part of the connection network of type B regulators, without presenting them as a separate topic. We have found important discussing them briefly to stress the highly developed range of dependencies in regulation in shoot induction and it seems inevitable to switch a bit among them (we have partially discussed it above). We do hope you will find it more accessible.

  1. In lines 364-365 the authors state that “in Different studies” it’s reported that de novo shot organogenesis engages different genetic programs from that of embryogenesis, but then they report only one reference. Where are the other studies referenced?

– the imprecise phrase has been changed and supported by the appropriate reference (line 363).

  1. What does the sentence in lines 389-91“although on the other CK biosynthetic gene LOG2 is up regulated in response to cytokinin (probably to help complement the phytohormone pool, perhaps to trigger the repression of CK biosynthesis)” mean? I think there is an error.

– it has been clarified. The cytokinin biosynthetic gene LOG2 is a primary target of type-B ARRs, that is upregulated in response to cytokinin. On the one hand, it potentially serves to help replenish the cytokinin pool, and probably to achieve cytokinin amount needed to trigger the repression of their excessive biosynthesis on the other (line 393).

  1. I would add a stand-alone paragraph for epigenetic reprogramming because now it is part of the “regulation of cytokinin-related genes in shoot development” section, and even if it relates also to the arguments discussed there because cytokinins control methylation marks on the WUS promoter, this section broadens a lot the topic presented in the title and doesn’t fit totally with it.

– the additional subsection has been entered (3.3.1 Epigenetic reprogramming in cytokinin signaling);

 Comments on the language:

  1. Throughout the manuscript, the language register sounds often colloquial (for example in line 224 “platform” to refer to a model organism) and generalist, as in line 211, where the authors should be more specific. Which “Adequate system” are the authors referring to? In line 214, “It is the most common device in prokaryotic organisms [27, 30].” Which common “device”?

– it has been corrected;

  1. In lines 285-288, besides the fact that the sentence sounds weird and the punctuation is wrong, what is a “recommendation signal”?

– it has been corrected; the subject of the review includes the cytokinin signal transduction. The sentence stresses the conversion/transformation of tis signal from a cytokinin molecule into a definite internal recommendation signal – a precise molecular/genetic instruction on further activity to be taken (line 409).

  1. In line 345, “CLAVATA3 (CLV3), also a transcription factor” is incorrect, and the authors themselves specify in the following lines that it is a small peptide.

– it has been corrected (line 342);

  1. In line 220 “cascade includes more (three) subfamilies” can be corrected with “cascade includes three subfamilies”

– it has been removed (line 235);

  1. In line 397 “Undoubtedly” should be removed, and so on

– the word “Undoubtedly” has been removed (line 449);

  1. The use of brackets in the manuscript is wide but not optimal. some sentences within brackets should be removed as they sound like authors’ personal considerations without any bibliographical reference, as in line 174 and in line 384; some others bear important information/assumptions that should be put outside the brackets and discussed, as in lines 202-203, in line 226, in lines272-273, in lines 336-338 and also in lines 358-359; in some other cases a reference should be added, as in line 225, and then discussed

– the required changes have been made.

There are typos throughout the ms:

  1. In line154-155 “Arabidopsis Histidine Kinase 4 (AHK 4 or Cytokinin Response 1), Arabidopsis Histidine Kinase 2 and 3” all the name of the genes must be written in capital letters – the manuscript has been checked for incorrectness, however in this case the name refers to the names of the receptor proteins (161);
  2. “nuclear” instead of “nucleus” in line 246 – it has been corrected (line 272);
  3. In line 258, “between cytoplasm and a nucleus” should be corrected in “the cytoplasm and the nucleus” – it has been corrected (line 287);
  4. “mediate” instead of “intermediate” in line 279 – it has been corrected (line 310);
  5. In the end of line 293, add “,” – it has been added (line 405);
  6. In line 343 write “WUS, a homeodomain transcription factor, is synthetized …” instead of “A WUS, homeodomain transcription factor is synthetized…” – it has been corrected (line 339);
  7. In line 350 “leucine-rich” instead of “leucine rich” – it has been corrected (line 348);
  8. In line 383, “cytokinin-rich” instead of “cytokinin rich” – it has been corrected (line 386);
  9. In line 440, “genes” instead of “gene” – it has been corrected (line 494);
  10. In line 580, add “about” between “studies” and “Cytokinin-“ – it has been corrected according to the native English speaker.

Sincerely yours,

Katarzyna Hnatuszko-Konka

Reviewer 2 Report

The review paper by Hnatuszko-Konka et al. entitled "The cytokinin signaling and de novo shoot organogenesis" summarizes some of the work done to date on the role of cytokinin (one of the essential plant phytohormones) and plant organ development with a focus on the shoot region.

Here are some recommendations:

Page 1, line 28 - remove keywords "cytokinin signaling and organogenesis" as they are already in the title

Page 2, line 55 - spelling of 'reprogramming'

Page 2, line 72 - DNSO can be used instead of writing out the full words (abbreviation was explained in line 69)

Page 2, line 84 - de novo in italics

Page 3, line 129- Sentence starting with "There is no doubt....." is ambiguous.

Page 4, line 155 - space missing between "3 are" and please check for lack of space between words and brackets etc throughout the manuscript.

Page 5, line 234 - verb "are" could be modified to "have been"

Page 6, line 250 - CHASE could be defined ie. extracytosolic sensing domain Cyclase/Histidine kinase Associated Sensory Extracellular

Page 6, line 286 to 287 - Modify the sentence from "And actually...."to "It is at this level that one....."

Page 7 to 14- one suggestion is to break section 3.3 Regulation of cytokinin-related genes in shoot development into subheadings/subsections. This could guide the readers on how cytokinin-related genes regulate shoot development  under small subheadings. For example on Page 13 starting with the sentence "Apart....SAM formation [56,57]" it seems to focus on chromatin, DNA methylation and epigenetics. It would be help with the flow by having subheadings. Use statement on Page 7 lines 292-295: "in fact, it seems that the CK-mediated de novo shoot development......(v)chromatin remodelers activity" as possible subheadings.

Page 7 line 297- the potential binding motifs for Type -B Response Regulators has been compiled. Do you mean for this review or that this has been done already. Reference missing if the latter.

Page 7, line 319 - do you mean hypotheses?

Page 7, line 322 - de novo in italics

Page 11, line 395 - "impossible in a way" could be removed from the sentence.

Page 12, line 470 - ARR10? or RR10?

Author Response

Responses to Reviewer #2

Dear Reviewer #2

I would like to thank you for the insightful revision. I appreciate all your comments and remarks.

Regarding your comments (present line numbers are given in brackets):

  1. Page 1, line 28 – remove keywords "cytokinin signaling and organogenesis" as they are already in the title – keywords have been modified;
  2. Page 2, line 55 – spelling of 'reprogramming' – it has been corrected;
  3. Page 2, line 72 (78) – DNSO can be used instead of writing out the full words (abbreviation was explained in line 69) – we have tried to avoid of using abbreviation at the beginning of the paragraph. We have not changed it, hoping you could accept it;
  4. Page 2, line 84 (90) – de novo in italics – it has been corrected;
  5. Page 3, line 129 (134) – Sentence starting with "There is no doubt....." is ambiguous – it has been clarified;
  6. Page 4, line 155 – space missing between "3 are" and please check for lack of space between words and brackets etc throughout the manuscript – it has been corrected;
  7. Page 5, line 234 (259) – verb "are" could be modified to "have been" – the sentence concerns current process of discovering, that is why we have used the given tense (the MS has been verified by the native speaker).
  8. Page 6, line 250 (278) – CHASE could be defined ie. extracytosolic sensing domain Cyclase/Histidine kinase Associated Sensory Extracellular – the CHASE abbreviation has been developed;
  9. Page 6, line 286 to 287 (413) – Modify the sentence from "And actually...."to "It is at this level that one....." – it has been modified;
  10. Page 7 to 14 – one suggestion is to break section 3.3 Regulation of cytokinin-related genes in shoot development into subheadings/subsections. This could guide the readers on how cytokinin-related genes regulate shoot development under small subheadings. For example on Page 13 starting with the sentence "Apart....SAM formation [56,57]" it seems to focus on chromatin, DNA methylation and epigenetics. It would be help with the flow by having subheadings. Use statement on Page 7 lines 292-295: "in fact, it seems that the CK-mediated de novo shoot development......(v) chromatin remodelers activity" as possible subheadings – section 3.3 has been rearranged and the additional subsection has been entered (3.3.1 Epigenetic reprogramming in cytokinin signaling);
  11. Page 7 line 297 (416) – the potential binding motifs for Type-B Response Regulators has been compiled. Do you mean for this review or that this has been done already. Reference missing if the latter. – it has been corrected and the missing reference has been added;
  12. Page 7, line 319 (440) – do you mean hypotheses? – it has been corrected;
  13. Page 7, line 322 (443) – de novo in italics – it has been corrected;
  14. Page 11, line 395 – to "impossible in a way" could be removed from the sentence – it was referring to “isolation” of the elements/genes engaged only to “pure” cytokinin-related shoot direct organogenesis; we have used this to place some emphasis for the unlikelihood of feasibility; if you find it acceptable, we would like to save it;
  15. Page 12, line 470 (528) – ARR10? or RR10? – it has been corrected;

Sincerely yours,

Katarzyna Hnatuszko-Konka

Reviewer 3 Report

This is a comprehensive review of the cytokinin signaling in de novo shoot organogenesis (DNSO). However, it a shame that there is not figure/cartoon to explain 1) the signal transduction of cytokinin and 2) the genetic network underlying cytokinin signaling during DNSO. The genes involved in shoot organogenesis is covered in P7 line 328–P12 line 481, but this is all rather stuffed in Table 1. A comprehensive diagram for understanding the relationship between these genes is lacking, we have a good review but it would be a shame if it were just a list of features. Indeed, the article would be stronger if the general introduction and conclusions were simplified (they are somewhat redundant and distract from the content regarding the cytokinin actions) and the above two figures were added.

Minor points:

P4, line 160

"They display structural similarity... weight of ca. 100 kDa."

When describing the structural similarity, it is better to show the amino acid sequence homology between AHK4 and AHK2/AHK3.

P4, line 187

After the sentence "...seem to be necessary", the observation of the crystal structure of AHK4-tZ complex should be presented (Hothorn et al., 2011). The AHK4-tZ complex reveals that the positions of the N3, N9, and N7 of the adenine ring buried in the binding pocket of AHK4. This observation supported the absence of hormonal activity of tZ ribosylated at the N9 position because the riboside moiety did not fit into the binding pocket.

Hothorn, M., Dabi, T. & Chory, J. Structural basis for cytokinin recognition by Arabidopsis thaliana histidine kinase 4. Nat. Chem. Biol. 7, 766–768 (2011).

P5, line 199

Replace "cis-zeatin > N6-benzyladenine" with "N6-benzyladenine > cis-zeatin".

Also, if you are referring to the structural differences of cytokinins (also in the conclusion), it would be better to provide the chemical structure of each cytokinin such as trans-zeatin, isopentenyladenine, and N6-benzyladenine.

P13, line 519

Typing error. Replace "METHYLOTRASFERASE 1" with " METHYLTRANSFERASE 1".

References, P15

The same reference is found in Reference No. 15 and No. 17.

Author Response

Responses to Reviewer #3

Dear Reviewer #3

I would like to thank you for the insightful revision. I appreciate your feedback.

Regarding your comments:

Language:

Following your suggestion the manuscript has been edited by a native English speaker (certificate attached).

  1. Indeed, the article would be stronger if the general introduction and conclusions were simplified (they are somewhat redundant and distract from the content regarding the cytokinin actions) and the above two figures were added”.

Diagrams:

  • The figure on the signal transduction of cytokinins with examples of genes involved has been added.
  1. Introduction and conclusions:

The general introduction and conclusions have been lightly amended. However, we find the information presented interesting and important. If you find it acceptable, we would like to save them. Please, note that, addressing your comments we have also tried to be consistent with opinions presented by two other reviewers, however we do hope we managed to explain all the issues.

  1. P4, line 160 (166) – "They display structural similarity... weight of ca. 100 kDa."

When describing the structural similarity, it is better to show the amino acid sequence homology between AHK4 and AHK2/AHK3 – it has been clarified (structural similarity was referring to domain similarity);

  1. P4, line 179 (187 – 191) – the information on the crystal structure of AHK4-tZ complex has been added.
  2. P5, line 199 (212) – Replace "cis-zeatin > N6-benzyladenine" with "N6-benzyladenine > cis-zeatin" – it has been clarified; we have based on publication by Lomin et al. (2015): “In particular, AHK3 was shown to have the highest affinity for trans-zeatin (tZ), much lower affinity for isopentenyladenine (iP), and the lowest affinity for cis-zeatin (cZ) and N6-benzyladenine (BA)”.
  3. Also, if you are referring to the structural differences of cytokinins (also in the conclusion), it would be better to provide the chemical structure of each cytokinin such as trans-zeatin, isopentenyladenine, and N6-benzyladenine – Figure 1 on the chemical structure has been added.
  4. P13, line 519 (578) – Typing error has been corrected.

  1. References:

The same reference is found in Reference No. 15 and No. 17 – improper references have been corrected.

Sincerely yours,

Katarzyna Hnatuszko-Konka

Round 2

Reviewer 3 Report

The authors have done a good job in revising the manuscript and have addressed the concerns and suggestions raised in my initial review. The figures have been added as well. I have nothing to add to this review, except that in line 211 on page 6. Based on a publication by Lomin et al. (2015), the affinity of benzyladenine (Kd =359 nM) for AHK3 is 4.5-fold greater than that of cis-Zeatin (Kd = 1602 nM). Please confirm Table 1 of Lomin's paper.

Author Response

Responses to Reviewer

Dear Reviewer,

I would like to thank you for the revision.

Regarding your comment:

The authors have done a good job in revising the manuscript and have addressed the concerns and suggestions raised in my initial review. The figures have been added as well. I have nothing to add to this review, except that in line 211 on page 6. Based on a publication by Lomin et al. (2015), the affinity of benzyladenine (Kd =359 nM) for AHK3 is 4.5-fold greater than that of cis-Zeatin (Kd = 1602 nM). Please confirm Table 1 of Lomin's paper.

– I have confirmed Table 1 of Lomin's paper. The order of affinity has been corrected (line 212).

Sincerely yours,

Katarzyna Hnatuszko-Konka
